# Retrospective development of a novel resilience indicator using existing cohort data: The adolescent to adult health resilience instrument

**Diana Montoya-Williams** [ID]*, **Molly Passarella, Scott A. Lorch**

Division of Neonatology, The Children's Hospital of Philadelphia, Philadelphia, PA, United States of America

* montoyawid@email.chop.edu

## Abstract

### Background

Cohort studies represent rich sources of data that can be used to link components of resilience to a variety of health-related outcomes. The Adolescent to Adult Health (Add Health) cohort study represents one of the largest data sets of the health and social context of adolescents transitioning into adulthood. It did not however use validated resilience scales in its data collection process. This study aimed to retrospectively create and validate a resilience indicator using existing data from the cohort to better understand the resilience of its participants.

### Methods

Questions asked of participants during one Add Health data collection time period (N = 15,701) were matched to items on a well-known and widely validated resilience scale called the Connor Davidson Resilience Scale. Factor analysis and psychometric analyses were used to refine and validate this novel Adolescent to Adult Health Resilience Instrument. Construct validity utilized participants' answers to the 10 item Center for Epidemiologic Studies Depression Scale, which has been used to validate other resilience scales.

### Results

Factor analysis yielded an instrument with 13 items that showed appropriate internal consistency statistics. Resilience scores in our study were normally distributed with no ceiling or floor effects. Our instrument had appropriate construct validity, negatively correlating to answers on the depression scale (r = -0.64, p<0.001). We also found demographic differences in mean resilience scores: lower resilience scores were seen among women and those who reported lower levels of education and household income.

**Data Availability Statement:** Parts of the datasets generated and/or analysed during the current study are publically available in the Adolescent to Adult Health repository (https://www.cpc.unc.edu/

projects/addhealth/documentation/publicdata).
However this study utilized the extensive restricted-
use data available by contractual agreement. Per
the Add Health website (https://data.cpc.unc.edu/
projects/2/view), "Restricted-Use Data will be
distributed only to certified researchers who
commit themselves to maintaining limited access.
To be eligible to enter into a contract, researchers
must complete Contract Application which
includes: Security plan IRB approval letter $1000
payment by check (NEW contract only). This
website also has links for how to apply for the
restricted-use dataset. No third party data was
used.

**Funding:** This research uses data from Add Health,
a program project directed by Kathleen Mullan
Harris and designed by J. Richard Udry, Peter S.
Bearman, and Kathleen Mullan Harris at the
University of North Carolina at Chapel Hill, and
funded by grant P01-HD31921 from the Eunice
Kennedy Shriver National Institute of Child Health
and Human Development, with cooperative funding
from 23 other federal agencies and foundations.
Special acknowledgment is due Ronald R.
Rindfuss and Barbara Entwisle for assistance in the
original design. Information on how to obtain the
Add Health data files is available on the Add Health
website (http://www.cpc.unc.edu/addhealth). No
direct support was received from grant P01-
HD31921 for this analysis.

**Competing interests:** The authors have declared
that no competing interests exist.

## Conclusions

It is possible to retrospectively construct a resilience indicator from existing cohort data and
achieve good psychometric properties. The Adolescent to Adult Health Resilience Instru-
ment can be used to better understand the relationship between resilience, social determi-
nants of health and health outcomes among young adults using existing data, much of
which is publicly available.

## Introduction

Resilience is defined as an individual's ability to positively adapt in the face of stress and/or
adversity so as to maintain in relatively stable or even good psychological and physical health
[1, 2]. Research on resilience and health has increased substantially in recent years due to a
shift in focus from studying health predominantly from a pathogenic orientation (i.e. what
causes disease) to exploring salutogenic forces (i.e. what promotes health) [3]. Individual psy-
chological resilience appears to be protective against a variety of physical and mental health
outcomes such as depression, anxiety and early mortality [4–6].

Resilience has been studied in a variety of ways: as a personality trait, a behavioral outcome
and at times, as a dynamic process that can be modulated [5]. Although one body of literature
has focused on resilience itself as the outcome [7–9], in other studies, investigators associated
health-related behaviors, disease risk factors or health outcomes themselves with different lev-
els of individual resilience [5, 6]. Still others have explored whether resilience moderates or
mediates outcomes for other conditions [10]. In the mental health field, for instance, some
researchers have sought to evaluate individual resilience as a mediator or moderator of treat-
ment uptake or success [11].

These diverse conceptualizations have led to a variety of ways in which resilience has been
operationalized and measured [12, 13]. Most studies looking at resilience and its relationship
to health and disease have prospectively evaluated individual resilience through the use of
resilience scales [4]. Although a wide range of scales exist, most of them are multi-dimensional,
measuring several constructs or themes [4, 12]. Some of the more commonly recurrent con-
structs that are believed to represent high resilience and thus are often included in validated
scales are a sense of personal agency, adaptive coping style, optimism and hopefulness and
social support [4, 14, 15]. Existing scales usually yield an estimation of resilience as a discrete
variable, with higher scores indicating higher resilience [12, 14, 16].

Given the costly and time-consuming nature of conducting prospective studies to assess
resilience in a population using a validated scale, some groups have utilized retrospective strat-
egies to assess individual resilience, by re-examining existing data from large cohorts. An
example of one large nationally representative cohort whose data has been utilized in this way
is the National Longitudinal Study of Adolescent to Adult Health (Add Health). The Add
Health cohort study is a nationally-representative longitudinal cohort study that recruited stu-
dents in the United States in grades 7–12 during the 1994–95 school year. After following ado-
lescents into adulthood with in-home surveys collected over five waves of data collection
spanning 24 years, it represents one of the broadest data sets on the social, economic, aca-
demic, psychological, and physical health status of adolescents transitioning into adulthood
[17].

Add Health explores both risk and protective factors for young adult health and achieve-
ment. Many of the questions asked of participants over the years relate to themes of self-

efficacy, optimism, persistence, social support, and faith/religiosity—themes that make up many validated resilience scales. However validated resilience scales were not used during any of the interviews. Over the years, researchers interested in the relationship between these themes and health have had to use alternate strategies to study resilience in this cohort [15, 18, 19]. Some research groups have written about resilience in the Add Health cohort as an adjective, exploring how or why individual or contextual factors may render some participants "resilient" to a pre-specified poor outcome, but without concretely measuring resilience itself [20–22]. Another angle has been the relationship between poor outcomes like depression, and individual concepts that are believed to represent one aspect of resiliency, such as personal agency [15], optimism [18] or social support [19]. In doing so, these researchers have utilized Add Health data to create a scale that operationalizes the specific resilience-related concept in question. For instance, Hitlin and Elder performed exploratory factor analysis using the first wave of data to construct a measurement model of agency [15] that has subsequently been used by others to measure the impact of agency on adolescent depression [18].

We built on this type of work by aiming to create an Add Health-based global resilience instrument that mirrored the structure of an existing widely used and well-validated resilience scale. The creation of such an instrument would allow the Add Health cohort study dataset to be used to more broadly examine resilience and its relationship to health outcomes, health behaviors, and social contexts.

## Methods

### Source of study sample

The present study was a cross-sectional analysis using the fourth wave of data collection from Add Health participants. Conducted in 2008, this wave consisted of in-home interviews of 15,701 young adults ages 24–32. It represented 80.3% of the participants initially recruited in the first wave of the Add Health study and per the original investigators, there were minimal differences between wave 1 and wave 4 responders [23]. Our study population included all young adults surveyed in this wave (N = 15,701).

Although some Add Health data is publicly available via the Add Health website (https://www.cpc.unc.edu/projects/addhealth/documentation/publicdata), this study utilized the extensive restricted-use data available by contractual agreement. This study was deemed exempt by our local institutional review board due to the publicly available de-identified nature of the Add Health data.

### Existing resilience scales

Although a more extensive review of existing resilience scales is out of the scope of this paper, a recent systematic review of resilience measurement scales by Windle et al. found that there is no current gold standard amongst the scales that have published psychometric validation data, in part because there is no gold standard for criterion validity for resilience [12].

Windle's group did conclude that the Connor-Davidson Resilience (CD-RISC) Scale, the Resilience Scale for Adults and the Brief Resilience Scale had the best psychometric ratings [12]. Of these, the CD-RISC has been validated in the broadest range of subjects of varying ages, ethnicities and preferred languages [8, 24, 25]. The CD-RISC was initially developed as a 25 item scale [16]. More recently, Campbell-Sills and Stein developed a 10 item version of the CD-RISC [26], which has also been validated in a variety of cross-cultural populations [27, 28]. Given the widespread use of the CD-RISC scales and the breadth of the existing validation literature for the CD-RISC 25 in particular, we chose the CD-RISC 25 as the model for the Add Health Resilience Instrument.

## Selection of candidate Add Health items

The original CD-RISC is composed of 5 major domains: personal competence or tenacity, strengthening effects of stress, positive acceptance of change, control, and spiritual influences [16]. In order to capture any Add Health data that might indicate resilience, all Add Health questions that reflected the items on the original expanded CD-RISC were pulled from the Add Health Wave 4 interview dataset. This led to 21 candidate Add Health items that were evaluated for inclusion in our Add Health Resilience Instrument (AHRI) (S1 Table). All of these items were scored on a Likert type scale.

## Exploratory factor analysis

An exploratory factor analysis of candidate Add Health items was conducted using a principal component analysis and a direct oblimin oblique rotation to allow for inter-item correlation. Eigenvalues > 1 were retained.

## Reliability

Internal consistency was evaluated by using Cronbach's alpha, where the recommended value ranges from 0.7 to 0.95 [29]. Items were eliminated if they lowered the overall instrument's alpha (i. the level of internal consistency between all the items) or if they did not load onto a factor well. We also assessed for internal consistency using item-test, item-rest and inter-item correlations. Item-test correlations determine how well each item correlates with the overall scale and should be roughly similar for all items [30]. The item-total correlation shows how the item correlates with a scale computed from only the other items; ideal values are above 0.2 [31]. Inter-item correlations identify items too similar or not similar enough in a scale, with recommended values between 0.15 and 0.5 [32].

## Instrument creation

After arriving at an instrument where items showed the most optimal internal consistency, items were reverse coded if negatively worded in order for higher scores to indicate higher resilience. Responses were reverse coded into a Likert type scale, with the highest score indicating a participant "strongly agreed" with a positive statement or strongly disagreed with a negative statement. There was no differential weighting for items; questions which asked participants to pick the frequency of feeling certain positive attributes over the past month had a maximum score of 3 while all other items had a maximum score of 2. This was done to better reflect the original potential answer choices as defined by Add Health investigators. Scores were then summed to create an overall AHRI score. The AHRI was intended to be used similarly to existing resilience scales, where scores are treated as a continuous variable and higher scores indicate higher resilience at the time of sampling. Ceiling and floor effects were analyzed by calculating the frequency of participants showing the minimum and maximum possible scores. Floor and ceiling effects of greater than 15% indicates limited content validity [29].

## Construct validity

As there are no gold standard criterion validity measures for resilience, we evaluated the AHRI through construct validity as others have done for the validation of other resilience scales [26, 33]. High levels of resilience are known to be protective against adverse mental health outcomes like depression [34, 35] or post-traumatic stress disorder [36]. Thus, we evaluated for convergent validity by calculating the correlation between AHRI scores and participants' scores on a depression scale that had been originally embedded in the fourth wave of Add

Health in home interviews. During the 2008 wave of data collection, participants completed the short form of the Centers for Epidemiologic Studies Depression Scale (CESD-10). Initially developed with 20 items [37], the CESD-10 scale is widely used for the identification and evaluation of depression in the general and adolescent populations [38, 39]. Various shorter forms of the 20-item CESD have been evaluated over the years, including the CESD-10 developed by Andresen et al. [40]. Using the existing CESD-10 data present in the original 2008 Add Health dataset, we were able to define a depression score for participants in our cohort. Possible responses for CESD-10 related items ranged from never/rarely (0) to most/all the time (3) resulting in a possible scale of 0–30. We categorized a score > 10 as indicative of adult depressive symptoms as has been recommended and done by others using this data for a similar purpose [41–43]. Resilience scores calculated using our AHRI were correlated with CESD-10 scores. Given the range and natural distribution of AHRI scores, we also created score tertiles to indicate low, medium or high resilience and compared CESD-10 scores in each of these three resilience categories via ANOVA analyses. Finally, we assessed for differences in AHRI scores between participants who had ever received a diagnosis of anxiety, depression or post-traumatic stress disorder in their lives and those who had not using Chi Square tests. Bonferroni correction was used to interpret results given the multiple tests done.

## Statistical analyses

Add Health oversampled certain subgroups by design, thus all analyses of this dataset required survey weighting in order for results to remain nationally representative [17]. Descriptive statistics were used to characterize AHRI scores first in the overall sample, then by gender and age. STATA, version 14, was used for all statistical analyses.

## Results

### Study population

Table 1 depicts the demographics of our cohort and overall resilience score distribution. The average age was 28.4 years (SD 1.9) and a little over half of participants were female (57%). The racial/ethnic makeup of the cohort was similar to that reported in the 2008 American Community Survey Data [44], which helped confirm the national representativeness of our cohort and verified that survey results were weighted appropriately. The majority of participants in our study (60.8%) finished high school but had not completed college. In addition, 46% of the cohort had a household income of less than $50,000 a year, which was slightly lower than the median household income reported by the Census Bureau's in 2008 [44].

### Psychometric analyses

Principal component analysis was conducted on the 21 candidate Add Health items that aligned with specific CD-RISC items (S1 Table). Items which lowered the scale's overall Cronbach's alpha were eliminated until we arrived at a 13-item AHRI with 3 factors that had an overall alpha of 0.78. Table 2 depicts the internal consistency metrics of each included item and the overall AHRI, with the last column demonstrating what the overall AHRI alpha would decrease to if that particular item were eliminated. Items retained the original wording used by the Add Health study investigators to enable future investigators to more easily utilize the AHRI using the publicly accessible Add Health codebook. The data in S1 Table lists the original variable names for ease of reference in the publicly available original Add Health codebook. Items that were eliminated were related to being stressed easily, the strength of a partner's commitment, the total number of close friends and religiosity (which made up part of the

**Table 1. Demographic characteristics (N = 15,701).**

| Variable | Mean (SD) or % of Cohort |
|---|---|
| **Age** (yrs) | 28.4 (1.9) |
| **Gender** (% Female) | 57.0% |
| **Race/Ethnicity** | |
| Non-Hispanic White | 65.6% |
| Non-Hispanic Black | 15.5% |
| Non-Hispanic American Indian | 2.3% |
| Non-Hispanic Asian | 3.4% |
| Other Non-Hispanics | 1.2% |
| Hispanic | 12.0% |
| **Highest Level of Education Achieved** | |
| Less than High School | 9.2% |
| Less than College | 60.8% |
| College Degree | 18.8% |
| More than College | 11.2% |
| **Household Income** | |
| <$20,000 | 12.7% |
| $20–49,999 | 33.0% |
| $50–99,999 | 39.4% |
| $100–149,999 | 10.2% |
| >$150,000 | 4.7% |

**Table 2. Factor loadings and internal consistency of each item, factor and overall Add Health Resilience Instrument (AHRI).**

| Factor | Item on AHRI | Variable Name in Add Health Data | Factor Loading | Item-test correlation[a] | Item-total Correlation[b] | Average inter-item correlation[c] | Alpha[d] |
|---|---|---|---|---|---|---|---|
| **Factor 1** | | | | | | | |
| | 1 | H4pe39 | .4379 | .5966 | .4771 | .1733 | .7560 |
| | 2 | H4pe37 | .7877 | .5079 | .4033 | .1862 | .7641 |
| | 3 | H4pe15 | .3522 | .5485 | .4328 | .1800 | .7608 |
| | 4 | H4pe38 | .7837 | .5369 | .4384 | .1845 | .7613 |
| | 5 | H4pe41 | .7658 | .5427 | .4595 | .1872 | .7613 |
| | 6 | H4pe33 | .5122 | .4453 | .3075 | .1883 | .7735 |
| **Factor 2** | | | | | | | |
| | 7 | H4mh4 | .5335 | .5532 | .4335 | .1787 | .7606 |
| | 8 | H4mh6 | .7254 | .6069 | .4836 | .1711 | .7551 |
| | 9 | H4mh2 | .6990 | .5070 | .3846 | .1835 | .7656 |
| | 10 | H4mh3 | .8376 | .6070 | .4781 | .1701 | .7558 |
| **Factor 3** | | | | | | | |
| | 11 | H4pe14 | .5885 | .4176 | .2732 | .1907 | .7773 |
| | 12 | H4pe23 | .6833 | .4939 | .3856 | .1870 | .7655 |
| | 13 | H4pe7 | .7878 | .4556 | .3334 | .1888 | .7702 |
| **Overall AHRI** | | | | | | **.1823** | **.7779** |

[a] Ideally these values are similar for all items.

[b] Recommended values are above 0.2.

[c] Recommended values are between 0.15 and 0.5.

[d] Ideal values for individual items are between 0.7 and 0.95.

**Table 3. Add Health Resilience Instrument.**

| Item | Max Score | Mean Score in Cohort | Mean Factor Score if Depressed |
|---|---|---|---|
| *Factor 1. Personal Competence* | **12** | **5.2** | **3.6** |
| 1. Q.There are many things that interfere with what I want to do.* | 2 | 0.6 | |
| 2. There is little I can do to change the important things in my life.* | 2 | 1.0 | |
| 3. I hardly ever expect things to go my way.* | 2 | 0.6 | |
| 4. Other people determine most of what I can and cannot do.* | 2 | 1.2 | |
| 5. There is really no way I can solve the problems I have.* | 2 | 1.1 | |
| 6. I go out of my way to avoid having to deal with problems in my life.* | 2 | 0.7 | |
| *Factor 2: Coping & Isolation* | **11** | **6.8** | **3.7** |
| 7. In the last 30 days, I have often felt confident in my ability to handle my personal problems. | 3 | 2.1 | |
| 8. In the last 30 days, I have often felt that difficulties were piling up so high that I could not overcome them.* | 3 | 1.8 | |
| 9. I often feel isolated from others.* | 2 | 1.1 | |
| 10. In the last 30 days, I have often felt unable to control the important things in my life.* | 3 | 1.8 | |
| *Factor 3. Optimism* | **6** | **2.4** | **1.6** |
| 11. I am not easily bothered by things. | 2 | 0.5 | |
| 12. Overall, I expect more good things to happen to me than bad. | 2 | 1.0 | |
| 13. I'm always optimistic about my future. | 2 | 0.9 | |
| **Instrument Totals** | **29** | **14.4** | **9.0** |

* These items were reverse coded in order for higher scores to indicate higher resilience.

extended 25 item CD RISC but not the refined 10 item version). The average inter-item correlation was 0.18 (Table 2).

We labeled the 3 factors within the AHRI as personal competence, coping/isolation, and optimism. Each item loaded predominantly on only one factor (Table 2). The maximum possible score on our AHRI was 29. Resilience scores in our study population using this constructed scale appeared normally distributed (S1 Fig). The skewness (0.01) and kurtosis (2.8) of the resilience scores were also consistent with a normal distribution. There were no ceiling or floor effects in our cohort (Less than 1% of participants scored either the minimum or maximum) [29]. Mean resilience score for this population was 14.4 (Table 3). Low AHRI scores were defined as < 10, medium as 10–19 and high as 20–29. The majority of the cohort (68%) fell into the medium resilience category.

In looking at differences in resilience by basic demographics, we found that the mean AHRI score was significantly lower among women compared to men (14.1 vs 14.6, p<0.001) (Table 4). In addition, resilience scores increased with increasing education and household income.

For the evaluation of our scale's convergent validity, we found a significant negative correlation between AHRI scores and CESD-10 scores for depression (r = -0.64, p<0.001). Tables 3 and 5 and S2 Fig all highlight that resilience scores increased as depression scores decreased. In addition, as seen in Table 5, the mean CESD-10 score for participants in the low Add Health resilience category was 11.6 and significantly different via pairwise comparisons to the mean

**Table 4. Differences in mean resilience scores by demographics.**

| Variable | Sample Size | Mean Resilience Scores (Standard deviation) | P-value |
|---|---|---|---|
| **Gender** | | | <0.001 |
| Female | 8352 | 14.1 (4.9) | |
| Male | 7349 | 14.6 (5.2) | |
| **Age** | | | 0.576 |
| Age < 30 | 10,534 | 14.4 (5.0) | |
| Age ≥ 30 | 5167 | 14.4 (5.4) | |
| **Race/Ethnicity** | | | 0.082 |
| Non-Hispanic White | 8266 | 14.5 (4.6) | |
| Non-Hispanic Black | 3341 | 14.2 (6.0) | |
| Non-Hispanic American Indian | 373 | 13.6 (5.5) | |
| Non-Hispanic Asian | 987 | 14.1 (7.0) | |
| Other Non-Hispanic | 190 | 14.6 (5.1) | |
| Hispanic | 2498 | 14.2 (5.7) | |
| **Highest Level of Education Achieved** | | | <0.001 |
| Less than High School | 1252 | 11.9 (4.1) | |
| Less than College | 9492 | 14.0 (5.0) | |
| College Degree | 3044 | 15.8 (4.9) | |
| More than College | 1909 | 16.2 (5.2) | |
| **Household Income** | | | <0.001 |
| <$20,000 | 1756 | 11.9 (4.7) | |
| $20–49,999 | 4801 | 13.9 (4.9) | |
| $50–99,999 | 5786 | 15.3 (4.9) | |
| $100–149,999 | 1563 | 16.2 (5.1) | |
| >$150,000 | 756 | 16.5 (5.3) | |

CESD-10 scores in the medium and high Add Health resilience categories (mean CESD-10 scores 5.5 and 2.6, respectively, Bonferroni corrected p-values <0.001). Over half of the participants in the low resilience category qualified for a diagnosis of depression per the CESD-10. Finally, there were significantly more participants who reported ever receiving a diagnosis of depression, anxiety or post-traumatic stress in the low AHRI score category than in high resilience categories (Bonferroni corrected p<0.001 for all).

## Discussion

In this study, we showed that data from the National Longitudinal Study of Adolescent to Adult Health could be used to create a global resilience indicator that demonstrated

**Table 5. Summary of construct validity findings.**

| Measure | Add Health Resilience Category | | | P-values |
|---|---|---|---|---|
| | Low (n = 2577) | Medium (n = 10,605) | High (n = 2519) | |
| **CESD-10 Score** | 11.6 (SD 4.9) | 5.5 (3.8) | 2.6 (2.6) | <0.001 |
| **Depressed per CESD-10** | 55.1% | 43.5% | 1.4% | <0.001 |
| **Ever received diagnosis of:** | | | | |
| Depression | 35.7% | 57.1% | 7.2% | <0.001 |
| Anxiety | 31.5% | 60.4% | 8.2% | <0.001 |
| Post-traumatic stress disorder | 38.8% | 54.2% | 7.0% | <0.001 |

appropriate psychometric properties among young adults. Using our indicator, we found that women had significantly lower resilience scores than men in this cohort. This has been documented by others noting gender differences in resilience using other scales [16, 28, 33]. We also found that resilience scores decreased as highest attained education level and household income decreased. This negative relationship between markers of socioeconomic status and resilience has been previously described [45–47], further validating our results.

Previous researchers had used Add Health data to explore the relationship between adolescent health and concepts related to resilience such as agency, self-efficacy or optimism [15, 18, 19]. Our work built on this existing literature in three important ways. We utilized data from when Add Health participants were firmly in their adult years (ages 24–32), creating the possibility of using the Add Health dataset to study resilience in adulthood in addition to adolescence. In addition, our study created an instrument that aims to capture a more global picture of resilience, rather than solely one aspect or factor. The three major factors we found in our constructed AHRI (personal competence, coping and isolation, and optimism) reflect major themes seen in prospectively validated resilience scales [14, 16], which speaks to the face validity of the AHRI and situates it within what has traditionally been believed to be important when assessing individual resilience. The main domain that exists on the CD RISC 25 which the AHRI does not capture well is the impact of spirituality on resilience. However, this domain was also dropped from the newer 10-item version of the CD RISC [26], with evidence of a more stable factor structure as a result. Thus, we did not feel the exclusion of spirituality to be a limitation of the AHRI per se. Finally, our construction of the AHRI shows that it is possible to take existing rich cohort data and re-purpose it for the identification of important constructs and outcomes beyond those conceived of during the design of the original study.

We do acknowledge some limitations. Given the existing nature of this data and the inability to re-contact participants, we were unable to assess the consistency of responses to our Add Health resilience scale over time, such as by looking at test-retest reliability. In addition, although Add Health has contacted the same participants over several waves, not every wave of interviews asks the same items and many of the resilience-related questions changed with each wave of interviews. Thus, participants' responses to the items we used to construct our resilience scale could not be assessed at any other time point using existing data. Finally, at the time of the fourth wave of sampling, Add Health participants were between the ages of 24–32 and as such, may not have had time to complete college or establish themselves in their careers. While this may have impacted some of the resilience score comparisons we conducted by education and household income, our findings of increasing resilience as level of education and household income bracket increased were consistent with trends others have previously documented [45, 47].

## Conclusions

The Add Health Resilience Instrument (AHRI) derived via interview data from a nationally representative longitudinal cohort study showed that a resilience indicator could be constructed retrospectively and evidence good psychometric properties. This instrument can be applied in future studies that utilize Add Health data to further explore the relationship between resilience, health, health behaviors and community context in young adults.

## Supporting information

**S1 Table. Characteristics of candidate Add Health items considered for inclusion in new instrument.**
(DOCX)

**S1 Fig. Distribution of Add Health Resilience Instrument scores.**
(TIF)

**S2 Fig. Evidence of the AHRI's construct validity using CESD-10 scores.**
(TIF)

## Acknowledgments

We would like to thank Ms. Ali Chandler for her assistance in editing and preparing this manuscript.

## Author Contributions

**Conceptualization:** Diana Montoya-Williams, Scott A. Lorch.

**Data curation:** Molly Passarella.

**Formal analysis:** Diana Montoya-Williams, Molly Passarella.

**Methodology:** Diana Montoya-Williams.

**Supervision:** Scott A. Lorch.

**Writing – original draft:** Diana Montoya-Williams.

**Writing – review & editing:** Diana Montoya-Williams, Molly Passarella, Scott A. Lorch.

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
