## [Decision Letter · Decision Letter 0]

4 Sep 2020

PONE-D-20-14860

Retrospective development of a novel resilience indicator using existing cohort data: the adolescent to adult health resilience instrument

PLOS ONE

Dear Dr. Montoya-Williams,

Thank you for submitting your manuscript to PLOS ONE. After careful consideration, we feel that it has merit but does not fully meet PLOS ONE’s publication criteria as it currently stands. Therefore, we invite you to submit a revised version of the manuscript that addresses the points raised during the review process.

Please respond to the two reviews point by point.  You will see that most of the issues the reviewers raise are concerned with clarifications and additional detail on the methods, sample, study design, and results of the study.  Both reviewers would like to see a more comprehensive and critical review of the literature on resilience.  I would add that you should focus this review on measures of resilience for adolescent and young adult populations.

We look forward to receiving your revised manuscript.

Kind regards,

Ellen L. Idler

Academic Editor

PLOS ONE

Journal Requirements:

3. We note you have included a table to which you do not refer in the text of your manuscript. Please ensure that you refer to Table 3 in your text; if accepted, production will need this reference to link the reader to the Table.

Reviewers' comments:

Reviewer's Responses to Questions

**Comments to the Author**

1. Is the manuscript technically sound, and do the data support the conclusions?

Reviewer #1: Partly

Reviewer #2: Partly

2. Has the statistical analysis been performed appropriately and rigorously? 

Reviewer #1: Yes

Reviewer #2: Yes

3. Have the authors made all data underlying the findings in their manuscript fully available?

Reviewer #1: Yes

Reviewer #2: No

4. Is the manuscript presented in an intelligible fashion and written in standard English?

Reviewer #1: Yes

Reviewer #2: Yes

5. Review Comments to the Author

Reviewer #1: Introduction

Please critically review previous instruments used to assess resilience.

Please add 21 candidate Add Health items that were evaluated for inclusion in AHRI as a supplement file.

Which method of factor rotation was used, why did you use this method, please add explanation into method section? Direct Oblimin Method? Promax Rotation?

The Methods section should be written as concisely as possible but should contain all elements necessary to allow interpretation and replication of the results. Such a sub titles something like these are recommended

The items & instruments

Study Samples (brief explanation about sampling in the fourth wave of original research)

Statistical analysis, Exploratory factor analysis, Known group comparison (construct validity), Reliability, Ethics

Please explain, is the scale is a screening one or a detective? Is there a cut-off point score for the scale?

Table 2, the last column is somewhat confusing. Did you indicate the alpha values for the scale if item deleted?

The alpha level of factor 3 is not satisfactory. What is the researcher explanation for such an alpha level?

Aren’t the data from 2008 too old to be presented as an original research?

Reviewer #2: 1. Is the manuscript technically sound, and do the data support the conclusions?

The proposed AHRI can be a useful tool for researchers working with the Add Health study. The manuscript is reasonably sound, and the evidence seems to support the conclusions the authors present. However, some technical matters require more attention and discussion.

I base the following six observations and recommendations on my comparison of the manuscript content with desiderata for factor analysis listed in page 94 of Bandalos’s and Finney’s 2010 “Factor Analysis: Exploratory and Confirmatory,” in Hancock’s and Mueller’s (editors) The Reviewer’s Guide to Quantitative Methods in the Social Sciences, published by Routledge.

1. The authors introduce the concept of resilience only very briefly. Although they mention typically used components of the latent construct, a fuller discussion of the theory behind or conceptual understanding of this construct would be helpful. For example, in the Discussion section, the authors mention that there is an ongoing debate about the best way to understand resilience: as a presumably fixed personality trait, as a behavioral outcome, or as a dynamic process. It stands to reason that the use of the concept may vary depending on different research questions, so the authors should make clear those research questions and the conceptual understanding or theory that links the concept of resilience to upstream predecessors or downstream outcomes. Making the conceptual understanding of resilience explicit will also help evaluate how well the candidate and selected Add Health items operationalize the concept.

2. Authors should indicate whether their factor analysis is exploratory or confirmatory (EFA, or CFA) and justify their decision, as each type of analysis serves a different purpose. The general guidance is to use the first type for newly developed constructs or when the theoretical basis is weak and to use CFA when the structure of the variable is well-established, as when using an independent sample to test a well-studied structure.

3. The authors should also indicate whether any of the variables selected as candidates for the factor analysis are not continuous. If so, they should note whether the variables are dichotomous or scales with less than five points. Such items may result in biased solutions and require a different type of analysis. Authors should identify all items, and including the variables’ codes will be helpful to researchers who wish to adopt the new instrument presented in this research paper. Also, the authors could have discussed whether the candidate items provide good-enough coverage of the dimensions of the latent concept. A table of the CD-RISC’s items, organized by domain, with the corresponding Add Health items lined up would be beneficial and illustrate whether any domains from the CD-RISC cannot be operationalized.

4. A discussion of the size of the analytical sample is missing: over 35 percent of the Wave IV sample was lost. Did one or a few items result in this loss of observations? If so, how does this affect the operationalization of the latent concept? Were non-responses concentrated among certain age, gender, racial or ethnic groups? If so, what impact does that have on the applicability of the proposed AHRI?

5. Difficulty factors that lead to misleading factor solutions may arise when variables with similar skewness and kurtosis can result in artefactual factors. To rule this out, or identify such factors, the authors should provide a summary of the candidate item’s descriptive statistics (mean, standard deviation, and values of skewness and kurtosis). Perhaps the table I proposed above can include CD-RISC items first, the corresponding candidate ASRI items next, then the percent missing responses for each item, and the descriptive statistics last.

6. I expected some discussion of the dimensionality of the chosen solution. The authors indicated that the CD-RISC operationalizes five domains, but the AHRI only has three. For example, the authors excluded the religiosity items from the selected solution, so the proposed AHRI does not operationalize the domain of spiritual influences. Is this not an important limitation of the proposed AHRI? Can the religiosity items from Add Health be retained to provide coverage for that domain, balancing a fuller operationalization with a weaker mathematical solution? What is the other domain not covered in the chosen solution, and can alternative solutions solve this?

The following seven observations concern my reading of the Methodology and Results sections:

7. I am not familiar with the evaluation of convergent validity. Based on its description (line 154 on), it seems to consist in evaluating the proposed AHRI through its correlation with a validated scale for depression, the CESD-10, which is empirically associated with extant resilience scales. I would appreciate a brief description of justification for this type of validation. Am I correct in thinking it consists of logically inverting the independent variable (resilience, which is theoretically upstream from the outcome of depression) with the dependent variable?

8. The second factor in the chosen factor solution, “Social Support / Feeling Overwhelmed,” is problematic. Having good social support and feeling overwhelmed strike me as related by not necessarily covarying concepts. A person who perceives the support of friends and family may feel overwhelmed by particularly stressful situations or traumatic events, while a person without support may feel overwhelmed more frequently.

9. I appreciated the analysis of average resilience scores for different demographic groups on page 11. However, given that some of the participants in the Add Health study may not have completed college, become financially independent, or had time to establish themselves in their careers *yet*, the comparisons by educational attainment and household income may not be the most meaningful.

10. Finally, I may be misinterpreting information on Tables 4 and 5 (on pages 11 and 12), but it seems that the confidence intervals for the scores of some of the comparison groups overlap. For example, at the top of Table 4, the mean AHRI score for Female is 13.8 with SD of 5.0, so the 95% confidence interval would run from around 3.8 to 23.8, roughly speaking, and would overlap the mean for Male, which is 14.4. Therefore, the means for Female and Male are not significantly different at the 95% confidence level, let alone at the 99.9% level.

2. Has the statistical analysis been performed appropriately and rigorously?

Yes, it has. However, as indicated above, some of the decisions the authors made require more discussion.

3. Have the authors made all data underlying the findings in their manuscript fully available?

No, the authors are contractually barred from sharing the restricted-use data from the Add Health study. However, Add Health makes a probability sample from the same wave the authors used available for public use. Confirming the results of the author’s analysis should be possible with that data.

4. Is the manuscript presented in an intelligible fashion and written in standard English?

Yes, but the authors should proofread for punctuation errors. They can also tighten the structure of the narrative to better guide the reader through their work.

Specific suggestions:

1. Some necessary commas are missing. For example, as written in line 116, the last domain of the CD-RISC seems to be, “control and spiritual influences,” when these are two distinct domains. I recommend the use of the Oxford comma, which adds clarity in lists. The authors may consider revising their manuscript to avoid using the passive voice, although this may be a field-specific preference.

2. I recommend the authors use the full name of the Add Health study in the first mention: National Longitudinal Study of Adolescent to Adult Health. The authors may be more familiar with cohort studies and retrospective analyses, but I consider it is more accurate to describe Add Health as a longitudinal study and to describe the author’s work as a cross-sectional analysis. It seems to be that a retrospective analysis would require predicting outcomes in Wave 4 (the wave from which the authors drew data) using independent variables from one of the earlier waves, but that is not the case. It would also be preferable if the authors distinguished the terms study and dataset (Add Health is a study, and several datasets are available for the various questionnaires that make up the study.) It is important to note that Add Health participants should not be described as patients, as being a patient was not a selection for inclusion.

3. Wave IV, conducted in 2008, is not the Add Health study’s original wave of data collection (line 155).

4. The description of the resilience scale developed by Hitlin and Elder previously belongs in the Introduction, together with other extant evidence is presented, rather than appearing towards the end of the paper.

6. PLOS authors have the option to publish the peer review history of their article (what does this mean?). If published, this will include your full peer review and any attached files.

Reviewer #1: **Yes: **Marzieh Araban

Reviewer #2: No

---

## [Author Response · Author response to Decision Letter 0]

5 Nov 2020

A Microsoft word formatted version of these responses have been attached.

Reviewer 1:

1. Please critically review previous instruments used to assess resilience.

Thank you for this comment. More information has been added to both the introduction and the methods. Although we utilized the Adolescent to Adult Health dataset, we have not focused on resilience scales specific to adolescents as suggested by the editor because our study sample was when participants in this cohort were firmly adults (ages 24-32). Thus the interpretation of our work and any potential future use of the instrument should be restricted to use within Add Health participants after they became adults. 

a. Inclusion of the following information in the introduction: “Resilience has been studied in a variety of ways: as a personality trait, a behavioral outcome and at times, as a dynamic process that can be modulated1 Most studies looking at resilience and its relationship to health and disease have prospectively evaluated individual resilience through the use of resilience scales.2 Although one body of literature has focused on resilience itself as the outcome,3–5, in other studies, investigators associated health-related behaviors, disease risk factors or health outcomes themselves with different levels of individual resilience.1,6 Still others have explored whether resilience moderates or mediates outcomes for other conditions.7 In the mental health field, for instance, some researchers have sought to evaluate individual resilience as a mediator or moderator of treatment uptake or success.8”

Location: Page 3, paragraph 2

b. “Existing scales yield an estimation of resilience as a continuous outcome, with higher scores indicating higher resilience.”

Location: Page 4, paragraph 1

c. The following paragraph was moved to the introduction: “Over the years, researchers interested in the relationship between these themes and health have had to use alternate strategies to study resilience in this cohort.[15,18,19] Some research groups have written about resilience in the Add Health cohort as an adjective, exploring how or why individual or contextual factors may render some participants “resilient” to a pre-specified poor outcome, but without concretely measuring resilience itself.[20–22] Another angle has been the relationship between poor outcomes like depression, and individual concepts that are believed to represent one aspect of resiliency, such as personal agency[15], optimism[18] or social support.[19] In doing so, these researchers have utilized Add Health data to create a scale that operationalizes the specific resilience-related concept in question. For instance, Hitlin and Elder performed exploratory factor analysis using the first wave of data to construct a measurement model of agency[15] that has subsequently been used by others to measure the impact of agency on adolescent depression.[18]”

Location: page 5, paragraph 1

d. We also discuss existing resilience instruments in one of the subsections of the Methods. We have relabeled this section “Existing Resilience Scales” for clarity and for ease of reference to readers. 

Location: page 6, paragraph 3

Given the existing strong review of resilience scale which we heavily relied on (Windle et al., 2011), we have added the following wording to the relabeled “Existing Resilience Scales” subsection of the Methods: “Although a more extensive review of existing resilience scales is out of the scope of this paper, a recent systematic review….” 

Location: page 6, paragraph 3

Finally, we believe it is important to justify why we picked the CD RISC as the model scale for our novel AHRI and thus revised the last paragraph of the “Existing Resilience Scales” subsection as follows: “Of these, the CD-RISC has been validated in the broadest range of subjects of varying ages, ethnicities and preferred languages.4,9,10”

Location: page 7, paragraph 1 

2. Please add 21 candidate Add Health items that were evaluated for inclusion in AHRI as a supplement file.

A new S1 Table has been created that lists all 21 candidate items along with other information requested by Reviewer 2 (Variable name in publicly available Add Health codebook, corresponding CD RISC factor, % missing as well as skewness and kurtosis of items included in factor analysis per Reviewer #2’s request). 

Location: Page 7, paragraph 1

3. Which method of factor rotation was used, why did you use this method, please add explanation into method section? Direct Oblimin Method? Promax Rotation?

We employed direct oblimin rotation. This has been added to the “Exploratory Factor Analysis subsection of the Methods.

Location: Page 8, Paragraph 1

4. The Methods section should be written as concisely as possible but should contain all elements necessary to allow interpretation and replication of the results. Such a sub titles something like these are recommended

a. The items & instruments, Study Samples (brief explanation about sampling in the fourth wave of original research), Statistical analysis, Exploratory factor analysis, Known group comparison (construct validity), Reliability, Ethics

We have revised the existing subtitles as follows on pages 4-8.

- Source of Study Sample

- Existing Resilience Scales

- Selection of Candidate Add Health Items 

- Exploratory Factor Analysis

- Reliability

- Instrument Creation

- Construct Validity

- Statistical Analyses

 Location: Pages 5-10

5. Please explain, is the scale a screening one or a detective? Is there a cut-off point score for the scale?

Thank you for this opportunity to clarify. Resilience scales typically indicate level of resilience on a continuum. The creators of existing scales have at times defined high, medium or low levels of resilience, such as the creators of Brief Resilience Scale,11 but for the most part scores are generated and analyzed as discrete variables in association studies. With regards to the CD-RISC scale, which was we used to create the AHRI, the creators did not specify a cut-off score; higher scores were meant to indicate higher resilience, though in initial validation studies, the creators did defined median scores and score quartiles from their samples.12 Subsequent literature has attempted to make associations between other conditions, variables or characteristics and different levels of resilience. For instance, patients with depression or anxiety have been shown to have lower CD RISC scores by some investigators.13 In short, resilience scales are not really conceptualized as screening or detective but rather indicative of the amount of resilience an individual might possess or be capable of at that moment. We conceptualized and developed the Add Health Resilience Instrument in the same way, whereby higher scores indicate higher levels of resilience at the time of the sampling. We did develop tertiles based on the distribution in our sample in order to allow for associations to be drawn between AHRI scores and CESD-10 depression scores provided by the original Add Health investigators for our construct validity analyses and to provide some information about AHRI scores by demographic characteristics as others have done. However, we did not conceptualize a specific cut-off score for the AHRI and envision future researchers using AHRI scores as a discrete count variable for future association studies.

We have added the following language to the “Instrument Creation” section of the Methods to indicate this intent: “The AHRI was intended to be used similarly to existing resilience scales, where scores are treated as discrete variables and higher scores indicate higher resilience at the time of sampling.”

Location: page 9, paragraph 1

We have also changed the language in the new “Construct validity” section of the Methods to better explain our approach in creating score tertiles. “Given the range and natural distribution of AHRI scores, we also created score tertiles to indicate low, medium or high resilience and compared CESD-10 scores in each of these three resilience categories via ANOVA analyses.”

Location: page 10, paragraph 1

6. Table 2, the last column is somewhat confusing. Did you indicate the alpha values for the scale if the item was deleted?

Thank you for this request to clarify. Yes, these alpha values depict how the overall alpha for the instrument would change if that particular item was dropped. In other words, would the overall cronbach’s alpha increase or decrease if we dropped that item? Items were excluded if eliminating them would increase the overall internal consistency of the AHRI and were retained if eliminating them would decrease the overall instrument’s alpha. We have added the following text to the “Psychometric analyses” subsection of the Results:

“Table 2 depicts the internal consistency and reliability metrics of each included item and the overall AHRI, with the last column demonstrating what the overall AHRI alpha would decrease to if that particular item were eliminated.

Location: page 12, paragraph 1

7. The alpha level of factor 3 is not satisfactory. What is the researcher's explanation for such an alpha level? 

While the alpha level for the combination of items that make up factor 3 is slightly lower than ideal values, all the items contained within Factor 3 contributed to a higher overall Cronbach’s alpha for the overall instrument and met a high threshold for alpha strength individually. In addition, there are references that suggest an alpha above 0.5-0.6 is acceptable in qualitative research.14 This comment, though, led us to consider whether it is necessary to include the alpha value for each factor, given that it may lead to confusion about the interpretation of alpha values. Thus we have eliminated the alphas for each sub-factor from Table 2 and just reported the alpha for each item and the overall AHRI alpha as that is traditionally what is reported in scale validation studies. If the reviewer and editors would like us to replace factor-level alphas, we are more than happy to put them back into Table 2. 

8. Aren’t the data from 2008 too old to be presented as an original research?

The originality of our research was in creating a novel instrument through the use of existing data so that this existing data can be explored by future researchers in novel ways. 

Reviewer 2: 

1. The authors introduce the concept of resilience only very briefly. Although they mention typically used components of the latent construct, a fuller discussion of the theory behind or conceptual understanding of this construct would be helpful. For example, in the Discussion section, the authors mention that there is an ongoing debate about the best way to understand resilience: as a presumably fixed personality trait, as a behavioral outcome, or as a dynamic process. It stands to reason that the use of the concept may vary depending on different research questions, so the authors should make clear those research questions and the conceptual understanding or theory that links the concept of resilience to upstream predecessors or downstream outcomes. Making the conceptual understanding of resilience explicit will also help evaluate how well the candidate and selected Add Health items operationalize the concept. 

Thank you for this suggestion. We have revised and expanded the introduction significantly to include a more in-depth conceptualization of resilience and how it has been previously studied. Specific attention was paid to how resilience is at times the outcome, sometimes the exposure and at other times the mediator or moderator in studies associating resilience with health behaviors, disease risk factors or outcomes. 

Location: page 3, paragraph 2 and 3 

2. Authors should indicate whether their factor analysis is exploratory or confirmatory (EFA, or CFA) and justify their decision, as each type of analysis serves a different purpose. The general guidance is to use the first type for newly developed constructs or when the theoretical basis is weak and to use CFA when the structure of the variable is well-established, as when using an independent sample to test a well-studied structure.

Thank you for this comment. This was an exploratory factor analysis as a resilience instrument utilizing Add Health items does not currently exist and we were developing a new instrument using existing data. We have clarified this in the Methods section by relabeling a subsection “Exploratory Factor Analysis.” It is conceivable that if more waves of data collection are conducted by Add Health researchers, they could utilize the work described in this paper to conduct an confirmatory factor analysis in a prospective fashion. 

Changes to the manuscript: The words “An exploratory factor analysis of candidate Add Health items…” was added to the new “Exploratory Factor Analysis” subsection of the Methods.

Location: page 7, last line and page 8, paragraph 1

3. The authors should also indicate whether any of the variables selected as candidates for the factor analysis are not continuous. If so, they should note whether the variables are dichotomous or scales with less than five points. Such items may result in biased solutions and require a different type of analysis. Authors should identify all items, and including the variables’ codes will be helpful to researchers who wish to adopt the new instrument presented in this research paper. Also, the authors could have discussed whether the candidate items provide good-enough coverage of the dimensions of the latent concept. A table of the CD-RISC’s items, organized by domain, with the corresponding Add Health items lined up would be beneficial and illustrate whether any domains from the CD-RISC cannot be operationalized.

All items that were included in the AHRI had likert type scales for the answers that had at least five points. This detail has been added to the Selection of Candidate Add Health Items subsection of the Methods:

Changes to the manuscript: “All of these items were scored on a likert type scale.”

Location: page 7, paragraph 2

We also made a new S1 Table with the 21 candidate Add Health items that were initially pulled from the fourth wave of questions utilized by the original Add Health investigators and indicated which 13 of these 21 items made the final cut into the AHRI. We have also added a column for the variable name within the Add Health Codebook that is publicly available online both in this supplementary table as well as in Table 3 in the main document for ease of reference. This new table also indicates which CD RISC factors we felt the candidate Add Health items mapped to. The domain within the original CD-RISC 25 that could not be operationalized within the AHRI was the one related to spiritual influences. We include a reference to this in our discussion, and also point out that the CD RISC 10 which is believed to be a stronger instrument with a more stable factor structure also dropped the spirituality domain of the original CD RISC 25. 

Changes to manuscript:“The main domain which the AHRI does not capture well is the impact of spirituality on resilience. However, this domain was also dropped from the newer 10-item version of the CD RISC,15 with evidence of a more stable factor structure as a result. Thus, we did not feel the exclusion of spirituality to be a limitation of the AHRI perse.”

Location: page 17, paragraph 1. 

4. A discussion of the size of the analytical sample is missing: over 35 percent of the Wave IV sample was lost. Did one or a few items result in this loss of observations? If so, how does this affect the operationalization of the latent concept? Were non-responses concentrated among certain age, gender, racial or ethnic groups? If so, what impact does that have on the applicability of the proposed AHRI? 

Thank you for this astute observation. Our initial analyses were run on a sample which had been restricted to those in wave 4 who were already parents based on future planned analyses. That is how 35% of the sample was lost. However, given this reviewer’s comment, we went back and re-did all of our analyses using the full cohort of 15,701 participants, as being a parent is not an absolute requirement for this scale. As the new S1 Table shows, data for each of the AHRI items was missing less than 1% of the time. Overall, there were minimal changes to the remainder of the results. Changes are summarized here:

- Differences between resilience scores in those below and over age 30 became non-significant (Table 1). The associated language describing that difference as significant has been removed from the results and the discussion.

- Small differences to the racial/ethnic, education and household income breakdowns reported in Table 1 such as the mean age of the cohort changing from 28.8 to 28.4 years for instance, the Hispanic percentage changing from 13% to 12% or the percentage of participants having earned a college degree changing from 11.6 to 18.8%

- No change to the items retained within the AHRI but one change to the factor structure. The item “I hardly ever expect things to go my way” moved to Factor 1.

- Minimal, non-significant change to the hundredth or thousandth decimal place for the numbers in Table 2. 

5. Difficulty factors that lead to misleading factor solutions may arise when variables with similar skewness and kurtosis can result in artifactual factors. To rule this out, or identify such factors, the authors should provide a summary of the candidate item’s descriptive statistics (mean, standard deviation, and values of skewness and kurtosis). Perhaps the table I proposed above can include CD-RISC items first, the corresponding candidate ASRI items next, then the percent missing responses for each item, and the descriptive statistics last. 

In the new S1 Table, we list the 21 candidate Add Health items, associated CD RISC 25 Factors and % missing responses. This new table also reports skewness and kurtosis data for the items that underwent factor analysis. None of those items had problematic skewness or kurtosis, using existing parameters in the literature.16 Mean scores for each AHRI item were already included in Table 3 where we believe they fit better, especially given the size of this new S1 Table. 

6. I expected some discussion of the dimensionality of the chosen solution. The authors indicated that the CD-RISC operationalizes five domains, but the AHRI only has three. For example, the authors excluded the religiosity items from the selected solution, so the proposed AHRI does not operationalize the domain of spiritual influences. Is this not an important limitation of the proposed AHRI? Can the religiosity items from Add Health be retained to provide coverage for that domain, balancing a fuller operationalization with a weaker mathematical solution? What is the other domain not covered in the chosen solution, and can alternative solutions solve this? 

Please see discussion above about the exclusion of the religiosity items in the AHRI and in more recent, perhaps more internally consistent and stable versions of the CD-RISC. We chose to model the AHRI off the CD RISC 25 vs. the CD RISC 10 in the hopes we would begin the exploratory factor analysis with the most inclusive sample of items possible. However, we feel that the resultant factor structure of the AHRI not only models the best internal consistency, but it also reflects the improved versions of the CD RISC and shows strong construct validity.

Changes to the manuscript: The following language was added to the discussion.

“The main domain that exists on the CD RISC 25 which the AHRI does not capture well is the impact of spirituality on resilience. However, this domain was also dropped from the newer 10-item version of the CD RISC,15 with evidence of a more stable factor structure as a result. Thus, we did not feel the exclusion of spirituality to be a limitation of the AHRI per se.”

Location: page 17, paragraph 1

7. I am not familiar with the evaluation of convergent validity. Based on its description (line 154 on), it seems to consist in evaluating the proposed AHRI through its correlation with a validated scale for depression, the CESD-10, which is empirically associated with extant resilience scales. I would appreciate a brief description of justification for this type of validation. Am I correct in thinking it consists of logically inverting the independent variable (resilience, which is theoretically upstream from the outcome of depression) with the dependent variable? 

That is correct. As there is no gold standard way to evaluate resilience, construct validity for many resilience scales relies on existing evidence of higher resilience negatively correlating with adverse mental health outcomes such as depression, anxiety or posttraumatic stress disorder. This is especially true given that it is unknown whether resilience is up or downstream of outcomes such as depression (or both). For instance, someone with lower resilience might be more likely to become depressed. However, given the documented modifiable nature of resilience, someone who develops depression may experience a decreased level of resilience than they might have evidenced previously. Given these ongoing conversations, we utilized the same convergent validity approach employed by other resilience researchers in previous scale validation work, looking for negative associations between resilience and adverse psychological or psychiatric outcomes like post-traumatic stress disorder9 and depression.15,17 

Location: This justification is described on page 9, paragraph 2, under the section labeled “Construct Validity.”

8. The second factor in the chosen factor solution, “Social Support / Feeling Overwhelmed,” is problematic. Having good social support and feeling overwhelmed strike me as related by not necessarily co-varying concepts. A person who perceives the support of friends and family may feel overwhelmed by particularly stressful situations or traumatic events, while a person without support may feel overwhelmed more frequently.

On looking at the items in Factor 2 again, we realize the phrase “social support” was not representative of the isolation item contained within factor 2 and thus have renamed Factor 2 “Coping/Isolation.” These two related topics are similar to Factor 3 in the original CD RISC 25 that was described as “related to positive acceptance of change and secure relationships.”12 

9. I appreciated the analysis of average resilience scores for different demographic groups on page 11. However, given that some of the participants in the Add Health study may not have completed college, become financially independent, or had time to establish themselves in their careers *yet*, the comparisons by educational attainment and household income may not be the most meaningful. 

This is a fair point and we have made a note in the limitations about this. We also included previous literature which found similar trends of increased resilience as education and income increased, which we believe lends some validity to our findings. 

Changes to the manuscript:

“Finally, at the time of the fourth wave of sampling, Add Health participants were between the ages of 24-32 and as such, may not have had time to complete college or establish themselves in their careers. While this have impacted some of the resilience score comparisons we conducted by education and household income, our findings of increasing resilience as level of education and household income bracket increased were consistent with trends others have previously documented.18,19

Location: Page 18, paragraph 1

10. Finally, I may be misinterpreting information on Tables 4 and 5 (on pages 11 and 12), but it seems that the confidence intervals for the scores of some of the comparison groups overlap. For example, at the top of Table 4, the mean AHRI score for Female is 13.8 with SD of 5.0, so the 95% confidence interval would run from around 3.8 to 23.8, roughly speaking, and would overlap the mean for Male, which is 14.4. Therefore, the means for Female and Male are not significantly different at the 95% confidence level, let alone at the 99.9% level.

This is a common misconception. The standard deviations presented (and the confidence intervals that can be constructed from them) are for estimations of each individual parameter (i.e. female score, male score, score for those < 30, etc.). They are not confidence intervals for the estimation of the difference between the groups. The only statistic we included that relates to the difference between groups is the p-value, which, in the gender example provided in the reviewer comment, is statistically significant to a p-value < 0.001. 

Specific suggestions: 

1. Some necessary commas are missing. For example, as written in line 116, the last domain of the CD-RISC seems to be, “control and spiritual influences,” when these are two distinct domains. I recommend the use of the Oxford comma, which adds clarity in lists. The authors may consider revising their manuscript to avoid using the passive voice, although this may be a field-specific preference. 

Oxford commas have been added throughout the manuscript. We will defer to the editors regarding the use of active vs. passive voice. 

2. I recommend the authors use the full name of the Add Health study in the first mention: National Longitudinal Study of Adolescent to Adult Health. The authors may be more familiar with cohort studies and retrospective analyses, but I consider it is more accurate to describe Add Health as a longitudinal study and to describe the author’s work as a cross-sectional analysis. It seems to be that a retrospective analysis would require predicting outcomes in Wave 4 (the wave from which the authors drew data) using independent variables from one of the earlier waves, but that is not the case. It would also be preferable if the authors distinguished the terms study and dataset (Add Health is a study, and several datasets are available for the various questionnaires that make up the study.) It is important to note that Add Health participants should not be described as patients, as being a patient was not a selection for inclusion. 

a. We agree with this reviewer’s conceptualization of our study as a cross-sectional analysis. However, we used the word retrospective in the title and manuscript to highlight the novel approach of creating a new instrument using data that has already been collected in a retrospective fashion. We worry that replacing the word retrospective, as we have employed it, with cross-sectional will mislead readers into wondering whether we collected data from Add Health participants ourselves. We wanted instead to emphasize that this was a retrospective approach in that we looked back and used existing data. We did double check to ensure that we did not ever describe our study as a retrospective study. Instead, each of the 5 instances in which that word was used was as an adjective or adverb, qualifying the construction of the instrument itself as retrospective. To ensure readers would not think our study design itself was a retrospective study as this reviewer describes, we have added this language to the first sentence of the Methods section:

 “The present study was a cross-sectional analysis using….”

 Location: page 6, paragraph 1

b. Thank you for your recommendation to include the full name of the Add Health study. The full name of the study was added to the introduction.

Location page 4, paragraph 2.

c. Thank you for catching our use of the word “patients” instead of “participants.” We have corrected this. Location, page 15, paragraph 1

3. Wave IV, conducted in 2008, is not the Add Health study’s original wave of data collection (line 155). 

The word “original” in this sentence has been deleted. 

Location, page 9, paragraph 2

4. The description of the resilience scale developed by Hitlin and Elder previously belongs in the Introduction, together with other extant evidence is presented, rather than appearing towards the end of the paper.

The agency scale created by Hitlin and Elder as well as the previous Add Health resilience-related research had been cited in the introduction in this sentence, “Researchers interested in the relationship between these themes and health have had to use alternate strategies to study resilience in this [Add Health] cohort, often focusing on single discrete concepts such as agency, self-efficacy, or optimism.20–22” However, given this reviewer’s excellent suggestion, we have re-organized the writing and expanded the description of previous Add Health-based resilience-related research in the introduction. 

Location: page 5, paragraph 1

References

1. Hu T, Zhang D, Wang J. A meta-analysis of the trait resilience and mental health. Pers Individ Dif. 2015;76:18-27.

2. MacLeod S, Musich S, Hawkins K, Alsgaard K, Wicker ER. The impact of resilience among older adults. Geriatr Nurs. 2016;37(4):266-272.

3. Cardoso JB, Thompson SJ. Common Themes of Resilience among Latino Immigrant Families: A Systematic Review of the Literature. Fam Soc. 2010;91(3):257-265.

4. Fu C, Leoutsakos J-M, Underwood C. An examination of resilience cross-culturally in child and adolescent survivors of the 2008 China earthquake using the Connor–Davidson Resilience Scale (CD-RISC). J Affect Disord. 2014;155:149-153.

5. Rosenberg AR, Bradford MC, McCauley E, et al. Promoting resilience in adolescents and young adults with cancer: Results from the PRISM randomized controlled trial. Cancer. 2018;124(19):3909-3917.

6. Topel ML, Kim JH, Mujahid MS, et al. Individual Characteristics of Resilience are Associated With Lower-Than-Expected Neighborhood Rates of Cardiovascular Disease in Blacks: Results From the Morehouse-Emory Cardiovascular (MECA) Center for Health Equity Study. J Am Heart Assoc. 2019;8(12):e011633.

7. Kishore MT, Satyanarayana V, Ananthanpillai ST, et al. Life events and depressive symptoms among pregnant women in India: Moderating role of resilience and social support. Int J Soc Psychiatry. 2018;64(6):570-577.

8. Kurtz SP, Pagano ME, Buttram ME, Ungar M. Brief interventions for young adults who use drugs: The moderating effects of resilience and trauma. J Subst Abuse Treat. 2019;101:18-24.

9. Green KT, Hayward LC, Williams AM, et al. Examining the factor structure of the Connor-Davidson Resilience Scale (CD-RISC) in a post-9/11 U.S. military veteran sample. Assessment. 2014;21(4):443-451.

10. Solano JPC, Bracher ESB, Faisal-Cury A, et al. Factor structure and psychometric properties of the Connor-Davidson resilience scale among Brazilian adult patients. Sao Paulo Med J. Published online May 13, 2016. doi:10.1590/1516-3180.2015.02290512

11. Smith BW, Dalen J, Wiggins K, Tooley E, Christopher P, Bernard J. The brief resilience scale: assessing the ability to bounce back. Int J Behav Med. 2008;15(3):194-200.

12. Connor KM, Davidson JRT. Development of a new resilience scale: the Connor-Davidson Resilience Scale (CD-RISC). Depress Anxiety. 2003;18(2):76-82.

13. Min J-A, Yu JJ, Lee C-U, Chae J-H. Cognitive emotion regulation strategies contributing to resilience in patients with depression and/or anxiety disorders. Compr Psychiatry. 2013;54(8):1190-1197.

14. Hair J, Black B, Babin BJ, Anderson RE. Multivariate Data Analysis, 7th Edition. Vol 7th ed. Pearson Education Limited; 2014.

15. Campbell-Sills L, Stein MB. Psychometric analysis and refinement of the Connor-davidson Resilience Scale (CD-RISC): Validation of a 10-item measure of resilience. J Trauma Stress. 2007;20(6):1019-1028.

16. Fabrigar LR, Wegener DT, MacCallum RC, Strahan EJ. Evaluating the use of exploratory factor analysis in psychological research. Psychol Methods. 1999;4(3):272-299.

17. Blanco V, Guisande MA, Sánchez MT, Otero P, Vázquez FL. Spanish validation of the 10-item Connor-Davidson Resilience Scale (CD-RISC 10) with non-professional caregivers. Aging Ment Health. Published online November 8, 2017:1-6.

18. Frankenberg E, Sikoki B, Sumantri C, Suriastini W, Thomas D. Education, Vulnerability, and Resilience after a Natural Disaster. Ecol Soc. 2013;18(2):16.

19. Reyes M-F, Satorres E, Meléndez JC. Resilience and Socioeconomic Status as Predictors of Life Satisfaction and Psychological Well-Being in Colombian Older Adults. J Appl Gerontol. 2020;39(3):269-276.

20. Williams AL, Merten MJ. Linking community, parenting, and depressive symptom trajectories: testing resilience models of adolescent agency based on race/ethnicity and gender. J Youth Adolesc. 2014;43(9):1563-1575.

21. Hitlin S, Elder GH. Agency: An Empirical Model of an Abstract Concept. Adv Life Course Res. 2006;11:33-67.

22. Wickrama KAS, Bryant CM. Community Context of Social Resources and Adolescent Mental Health. J Marriage and Family. 2003;65(4):850-866.

---

## [Decision Letter · Decision Letter 1]

24 Nov 2020

Retrospective development of a novel resilience indicator using existing cohort data: the Adolescent to Adult Health Resilience Instrument

PONE-D-20-14860R1

Dear Dr. Montoya-Williams,

We’re pleased to inform you that your manuscript has been judged scientifically suitable for publication and will be formally accepted for publication once it meets all outstanding technical requirements.  Your attention to reviewers' comments and your revision were careful and thorough.

Kind regards,

Ellen L. Idler

Academic Editor

PLOS ONE

Additional Editor Comments (optional):

Reviewers' comments:

Reviewer's Responses to Questions

**Comments to the Author**

1. If the authors have adequately addressed your comments raised in a previous round of review and you feel that this manuscript is now acceptable for publication, you may indicate that here to bypass the “Comments to the Author” section, enter your conflict of interest statement in the “Confidential to Editor” section, and submit your "Accept" recommendation.

Reviewer #2: All comments have been addressed

2. Is the manuscript technically sound, and do the data support the conclusions?

Reviewer #2: Yes

3. Has the statistical analysis been performed appropriately and rigorously? 

Reviewer #2: Yes

4. Have the authors made all data underlying the findings in their manuscript fully available?

Reviewer #2: Yes

5. Is the manuscript presented in an intelligible fashion and written in standard English?

Reviewer #2: Yes

6. Review Comments to the Author

Reviewer #2: This is carefully conducted and well explained research that is interesting work in its own right. It may also serve as a methodological example for other researchers who need to adapted existing existing psychometric scales to the Add Health study and then validated the adapted scales. Thanks for addressing the comments from the first review; I hope some of them were helpful.

7. PLOS authors have the option to publish the peer review history of their article (what does this mean?). If published, this will include your full peer review and any attached files.

Reviewer #2: No

---

## [Editor Report · Acceptance letter]

2 Dec 2020

PONE-D-20-14860R1 

Retrospective Development of a Novel Resilience Indicator using Existing Cohort Data: The Adolescent to Adult Health Resilience Instrument  

Dear Dr. Montoya-Williams:

I'm pleased to inform you that your manuscript has been deemed suitable for publication in PLOS ONE. Congratulations! Your manuscript is now with our production department. 

Kind regards, 

on behalf of

Professor Ellen L. Idler 

Academic Editor

PLOS ONE